# The Effect of the Angle of Pipe Inclination on the Average Size and Velocity of Gas Bubbles Injected from a Capillary into a Liquid

Anastasia E. Gorelikova [1,2,*], Vyacheslav V. Randin [1,2,†], Alexander V. Chinak [1] and Oleg N. Kashinsky [1]

1 Heat Mass Transfer Problem Laboratory Power Engineering, Kutateladze Institute of Thermophysics SB RAS, 630090 Novosibirsk, Russia
2 Department of Physics, Novosibirsk State University, 630090 Novosibirsk, Russia
* Correspondence: gorelikova.a@gmail.com
† Deceased.

**Abstract:** This work is devoted to an experimental study of the effect of coalescence on the average diameter and velocity of gas bubbles in an inclined pipe. The measurements were carried out for agas flow rate of 3.3 and 5 mL/min at pipe inclination angles of 30–60°. The study of gas bubble diameters was performed using a shadow photography method. The values of the average diameter and velocity of the bubbles were obtained depending on the angle of inclination of the pipe. A map of regime parameters was constructed at which gas bubbles form a stable structure—a chain of bubbles with an equal diameter.

**Keywords:** inclined pipe; bubbles; coalescence; bubble velocity; chain of bubbles

## 1. Introduction

Gas bubbles floating in a stationary or moving liquid can have a significant effect on heat exchange due to the mixing of the wall layers of the liquid. The paper [1] shows that, in a downward bubbly flow, a change in the size of the dispersed phase can lead to both intensification and deterioration of the heat transfer as compared to a single-phase flow at constant rates of liquid and gas flow at the channel inlet. Adding small gas bubbles to the flow leads to "laminarization" in the wall region and deterioration in the heat transfer by about 25% as compared to a single-phase flow. Large bubbles lead to a higher turbulence level in the near-wall region, an increase in the average friction, and an intensification of the heat transfer up to 50%. In the paper [2], the addition of air bubbles led to a significant increase in the heat transfer rate (up to 300%) in a downstream bubbly flow in a sudden pipe expansion.

The dynamics of bubbles can be nonlinear and complex [3]. The movement of bubbles is influenced by the geometry of the channel, the internal diameter of the capillary on which the bubble is formed [4], and the characteristics of the liquid [5].

The paper [6] presented an experimental investigation of highly deformed bubbles. The dynamics of bubble formation were investigated via experiments with ultrapure water and silicone fluids, with gas flow rates of 5–300 mL/min. The behavior of bubbles in the immediate vicinity of the capillary was considered. It was shown that the coalescing dynamics are affected by capillary inertia; the viscosity has a negligible effect on coalescence. A phase diagram for coaxial coalescence and no coalescence was presented in terms of the Weber number and the Morton number to describe the effects of inertia and fluid properties on the dynamics of bubble formation.

Much attention has been paid to the study of vertical pipes and channels [7–9]. The efficiency of heat and mass transfer and gas-phase residence time largely depends on the dynamics of gas bubbles. According to the paper [10], gas–liquid interaction in the bubble

flow can have a significant impact on the size and velocity of bubbles, the area of interaction between gas and liquid, and the number of small bubbles.

Studies of multiphase systems with horizontal channels have been widely presented [11,12]. In the paper [13], an experimental investigation of bubbly flow in an annulus pipe was presented. It was shown that the presence of various values of total dissolved solid materials can affect the thermal conductivity of water and the bubble formation characteristics.

Much less attention has been paid to inclined pipes and channels, despite the fact that the angle of inclination can make a significant contribution to the nature of gas–liquid flows and affect the bubbles that move in stationary liquid.

The paper [14] presented an experimental study of the velocity of a free-floating gas slug in glass tubes with diameters of 11.8–30 mm using a time-of-flight method combined with high-speed video shooting and numerical processing of sequential images. The presented results indicate that the bubble velocity depends on the angle of inclination of the pipe and is non-monotonic in nature.

The paper [15] presented the results of an experimental study of the rise of single bubbles in an inclined flat channel. The fluid velocity was 0–0.2 m/s, the volume of one bubble varied in the range from 1 to 80 mL, and the channel depth was 8, 4 or 1.5 mm. It was shown that with the vertical arrangement of the channel, the bubble velocity primarily depends on the depth of the channel. When the angle of inclination of the channel changes from 0° to 90°, the velocity of the bubble monotonously increases and reaches a maximum with the vertical arrangement of the channel. This differs from the results obtained for projectile flows in round inclined pipes with large cross sections, where the velocity reaches a maximum at angles of inclination close to 45°.

The paper [16] studied the dynamics of growth and collapse of vapor bubbles for a laser-induced bubble on or near the wall. The difference in the shapes of bubbles near and on the wall was demonstrated. Deformation of a spherical bubble floating near the wall was noted. It was shown that the lifetime of the bubble near the wall was longer than the lifetime of the bubble on the wall.

In [17], liquid bubbles with a relative spherical diameter in the range of 4.75–9.14 mm floating near an inclined surface (inclination angle of 30°) were investigated. The authors discuss an increase by up to 8 times in the local heat transfer coefficient in comparison to the case of free convection, in which the average heat transfer value increases by 2 times compared to in the case without the addition of the gas phase.

In the paper [18], a study of three-dimensional trajectories of bubbles in a stationary liquid with working volumes of $300 \times 300 \times 1500$ mm$^3$ and $300 \times 150 \times 500$ mm$^3$ was carried out. Bubbles with a diameter of 3–5 mm were considered (depending on the capillary and fluid flow, the diameter of the bubbles changed slightly). The dependences of the effect of gas flow rate, height of the liquid column, and nozzle diameter on bubble ascent trajectory and deformation of the bubble surface are shown.

The paper [19] presented a bubble coalescence model consisting of two steps. First, an existing model of coalescence of bubbles of the same diameter in a turbulent flow was expanded to the case of bubbles of different diameters. In the second step, the obtained expression for the coalescence rate is used to obtain the dependences of the kinetic equations on the bubble diameter, which can be estimated using CFD packages (CFD, Computational Fluid Dynamics). As a result, a compact expression is obtained to describe the evolution of bubble sizes.

The paper [20] is devoted to the development of three-dimensional models of multiphase models for calculations and analysis of processes in nuclear reactors (Pressurized Water Nuclear Reactor (PWR)). A wide overview of methods is given, including various boiling models, Direct Numerical Simulation (DNS), calculations of adiabatic flows, etc. Despite the fact that models have been developed since 1980, and significant progress has been made in forecasting flows, the authors point out the need for further improvement in methods and for comparison of the results obtained with experimental data.

It is impossible not to note the convenience and relatively low cost of computer modeling of two-phase flows in comparison to experimental works. However, when using various software packages, the question of the applicability of the algorithms, validation of the data obtained, and appropriateness of the models developed for calculating the physical quantities of gas–liquid flows becomes acute. In addition, due to the complexity of the flow structure, it is often difficult to obtain results without using already existing empirical data.

Despite the increasing interest in the influence of the angle of inclination on the movement of bubbles and on gas–liquid flows, there is a significant shortage of experimental data. In the empirical work, attention is mostly paid to large gas consumption and large-diameter bubbles, and angles of inclination of the channels close to the vertical or horizontal position are chosen.

This work is devoted to the experimental study of the effect of coalescence on the average size and velocity of gas bubbles in an inclined pipe. The values of the average diameter and average velocity of the bubbles were obtained depending on the angle of inclination of the pipe. A map of regime parameters was constructed at which gas bubbles form a stable structure—a chain of bubbles with an equal diameter.

## 2. Experimental Setup and Technique

The scheme of the experimental setup is shown in Figure 1. Gas (air) was supplied from the compressor through the gas flow meter (1) and introduced into the liquid through one capillary of 0.2 mm inner diameter (2). The gas flow rate was monitored using a mass flow controller, Aalborg GFC17 (flow range 0–10 mL/min, accuracy ± 0.1 mL/min). The test section was a 1.2 m long tube made of Plexiglas with an inner diameter of 32 mm. Bubbles were filmed by a Nikon Zfc camera (4) through an optical section, and the flow was illuminated by an LED matrix (5). The distance from the location of the gas-phase introduction to the measurement point was from 100 to 600 mm. The channel inclination angle $\theta$ was counted from the vertical line, with the value $\theta = 0°$ corresponding to the vertical position of the channel and $\theta = 90°$ to the horizontal position. The measurements were carried out for the gas flow rates of 3.3 and 5.0 mL/min at pipe inclination angles of 30–60°. This range was chosen because, at similar channel inclination angles, there was a significant increase in heat exchange from the channel wall for gas–liquid bubble flows [21].

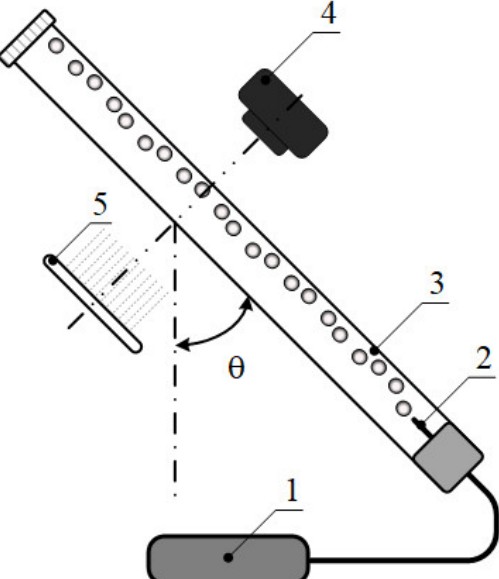

**Figure 1.** Experimental setup.: 1—flow rate meter; 2—capillary; 3—test section; 4—camera; 5—LED matrix.

Distilled water was used as the working fluid in the experiments, and air was used as the working gas. The exact physical properties of the working fluid [22] are presented in Table 1. All experiments were carried out at the standard pressure (1 atm) and room temperature (20 °C ± 1).

**Table 1.** Physical properties of the liquid used in this study (at 20 °C).

| Fluid | Density, $\varrho$ (kg/m$^3$) | Viscosity, $\mu$ (mPa·s) | Surface Tension, $\sigma$ (N/m) |
|---|---|---|---|
| Distilled water | 998 | 1 | 0.072 |

Figure 2 shows an example of the software processing of the received images. The processing was carried out in two stages. In the first stage, the videos were split into separate frames and translated into the "Grayscale" format. According to the level of illumination and the gradient illumination, the boundaries of the objects were found, and the frames were binarized. In the second stage of processing, the properties of the objects—size and location—were found. Correlation analysis of successive frames was carried out to determine the speed and trajectories of the bubbles.

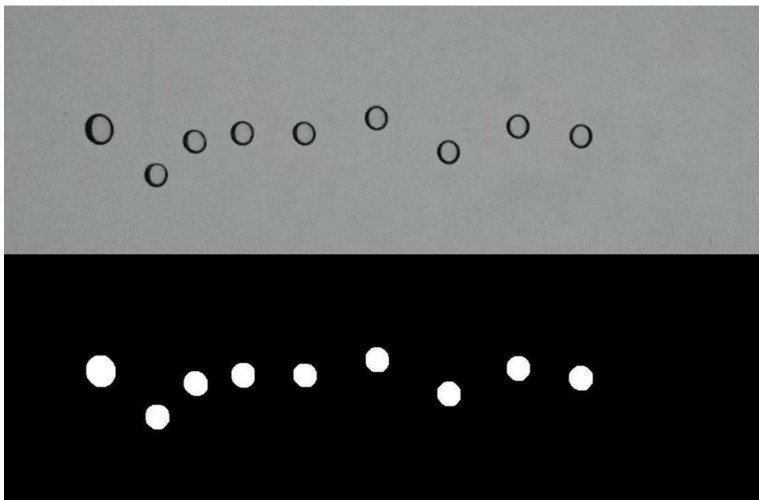

**Figure 2.** Example of image preprocessing for calculating the diameter and velocity of gas bubbles.

The resulting images were processed numerically, similarly to the method described in [23]. The diameter of gas bubbles was calculated from the area of the bubble in the image as an equivalent diameter according to the formula:

$$D = \sqrt{4S/\pi} \tag{1}$$

where $S$ is the area of the bubble in the image.

The size of the main part of the bubbles in the experiments did not exceed 2–3 mm. Bubbles of this size are close in shape to a sphere, which allows the use of this approximation.

The absolute accuracy of determining the diameter of bubbles $\Delta D$ is calculated using the formula:

$$\Delta D = \sqrt{\left(\frac{\partial f}{\partial S}\Delta S\right)^2} \tag{2}$$

Using the formula for determining the equivalent diameter of the bubble (1) as a function in Equation (2), we obtain an expression for the relative accuracy of determining the diameter $\delta D$:

$$\Delta D = \sqrt{\left(\frac{\partial f}{\partial S}\Delta S\right)^2} = \frac{\partial\left(\sqrt{\frac{4S}{\pi}}\right)}{\partial S}\Delta S = \frac{1}{2}\sqrt{\frac{4}{\pi S}}\Delta S \frac{1}{2}\sqrt{\frac{4S}{\pi S^2}}\Delta S = \frac{1}{2}\sqrt{\frac{4S}{\pi}}\frac{\Delta S}{S} = \frac{1}{2}D\delta S \quad (3)$$

$$\frac{\Delta D}{D} = \delta D = \frac{1}{2}\delta S \tag{4}$$

The accuracy of determining the position of the bubble boundary was $\pm 1$ pixel. According to the calibration frames, 1 mm is equal to 22 pixels, and the relative accuracy of determining the diameter for bubbles of 0.3–7 mm is 0.01–0.1. The value of the gas flow rate obtained during video processing converged with the values of the mass flow controller with an accuracy of 0.05.

The velocity of the bubbles was calculated from the frame-by-frame shift.

To validate the measuring system, a series of experiments were carried out with a vertical orientation of the pipe ($\theta = 0°$) and a gas flow rate $Q_g = 3.3$ mL/min.

The rate of bubble ascent is influenced by the buoyancy force

$$F_B = \frac{\pi D_b^3}{6}(\rho_L - \rho_G)g \tag{5}$$

and the drag force

$$F_D = \frac{\pi D_b^2}{4}C_d\frac{\rho_L}{2}V_b^2 \tag{6}$$

For a large volume of a quiescent liquid, the bubble rise velocity is

$$V_b = \sqrt{\frac{4D_b(\rho_L - \rho_G)g}{3\rho_L C_d}} \tag{7}$$

where $C_d$ is the drag coefficient. Bubbles with a small diameter (up to 3–4 mm) do not experience pulsation of shape, and their movement can be described as the movement of a rigid sphere. When liquid flows around a rigid sphere, the drag coefficient is equal to [24]:

$$\begin{cases} C_d = 24/Re_b, & at\ Re_b < 2 \\ C_d = 18.5/(Re_b)^{0.6}, & at\ 2 < Re_b < 500 \\ C_d = 0.44, & at\ 500 \le Re_b \end{cases} \tag{8}$$

where the Reynolds number $Re_b$ is calculated relative to the velocity $V_b$ and diameter $D_b$ of the gas bubble. Measuring values were compared with the calculated bubble rise velocity for the same bubble diameters (Figure 3).

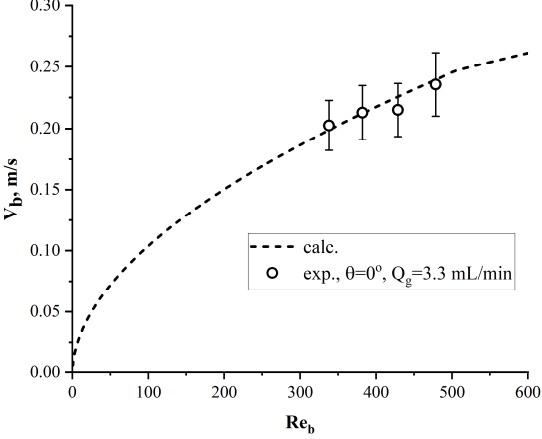

**Figure 3.** Comparisons of bubble rise velocity in vertical pipe ($\theta = 0°$) in experiments, and calculated bubble rise velocity.

## 3. Results and Discussion

### 3.1. Bubble Chain Movement Mode

Figure 4 shows characteristic images of bubbles with equal gas flow rates ($Q_g$ = 3.3 mL/min) and equal distances from the gas input point to the shooting point (*L* = 200 mm) for different angles of inclination of the pipe ($\theta$ = 30°–60°).

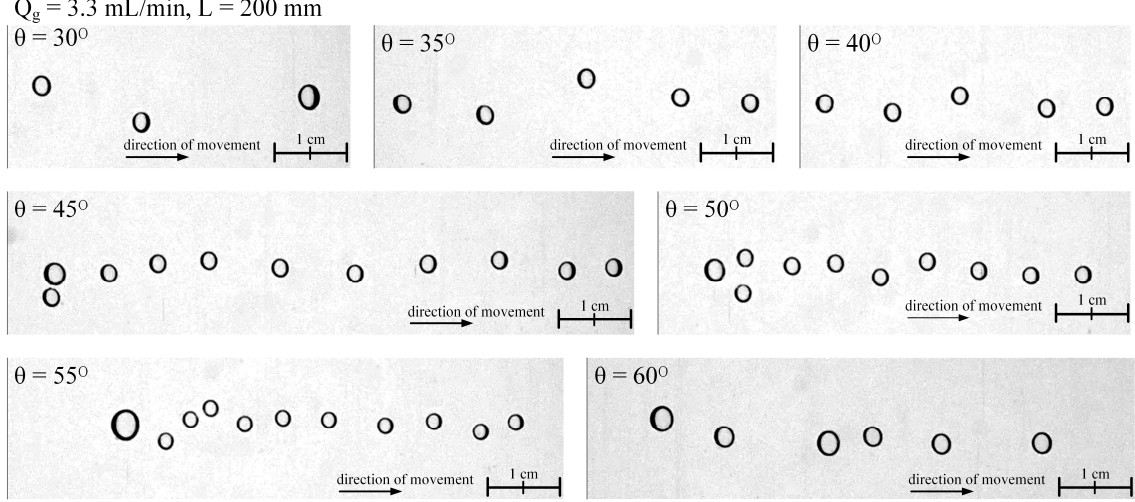

**Figure 4.** Characteristic images of bubbles. Gas flow rates $Q_g$ = 3.3 mL/min; distances from the gas input point to the shooting point L = 200 mm; angle of inclination of the pipe $\theta$ = 30°–60°.

At channel inclination angles $\theta$ = 30°–35°, the diameter and the nature of the movement of bubbles largely depended on their departure diameter and coalescence near the capillary. The bubbles oscillated perpendicular to the direction of motion and moved one at a time or in small, unstructured groups.

At pipe inclination angles from 40° to 55°, the bubbles formed a chain. The perpendicular velocity component was significantly reduced due to friction against the upper wall of the pipe. Between 5 and 15 bubbles in the chain had the same diameter, and the last one had a significantly increased diameter in comparison with the others. The long chains of bubbles were formed at angles of inclination $\theta$ = 45°–50°. With a further increase in the angle of inclination of the pipe, the distance between the bubbles decreased. Reducing the distance between the bubbles led to the coalescence of the bubbles. This resulted in the destruction of the chain of bubbles at an angle $\theta$ = 60°.

Figure 5 shows an example of bubble movement in modes without cluster formation (pipe inclination angle $\theta$ = 35°) and with cluster formation (pipe inclination angle $\theta$ = 50°) at a gas flow rate $Q_g$ = 3.3 mL/min. For cluster modes of bubble movement, the average values of bubble size and velocity were characteristically close to the values of size and velocity for a single bubble in a chain (deviation of no more than 5%). For modes without cluster formation, there were several typical bubble diameters that moved with different velocities.

Figure 6 shows a map of the modes of movement of the bubbles, as well as the area of formation of a chain of bubbles.

For gas flow rates from 3.0 to 5.0 mL/min and distances from the gas input point to the shooting point from 100 to 600 mm at pipe inclination angles of 30°–35°, the formation of a chain of bubbles did not occur. The oscillatory movements of the bubbles did not allow them to form a stable cluster structure.

$Q_g$ = 3.3 mL/min, L = 200-300 mm

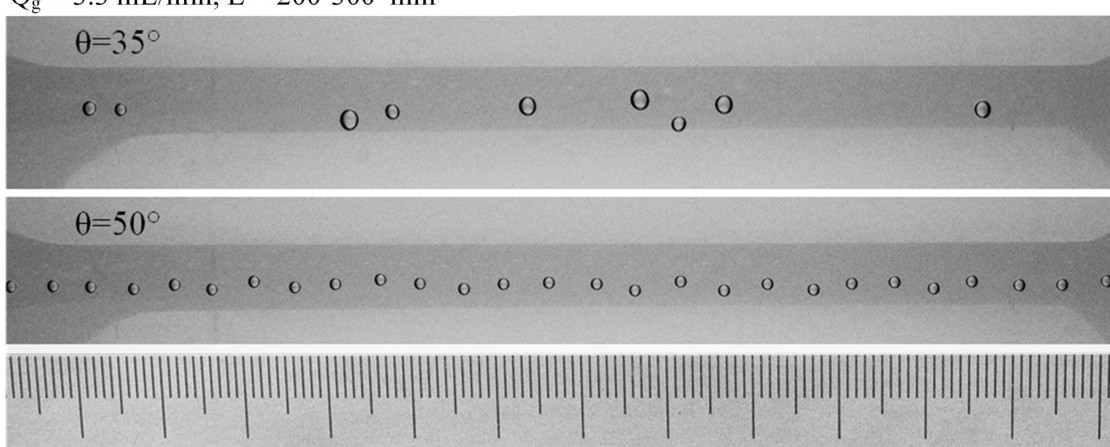

**Figure 5.** Bubble movement in modes without cluster formation (θ = 35°) and with cluster formation (θ = 50°); gas flow rate $Q_g$ = 3.3 mL/min.

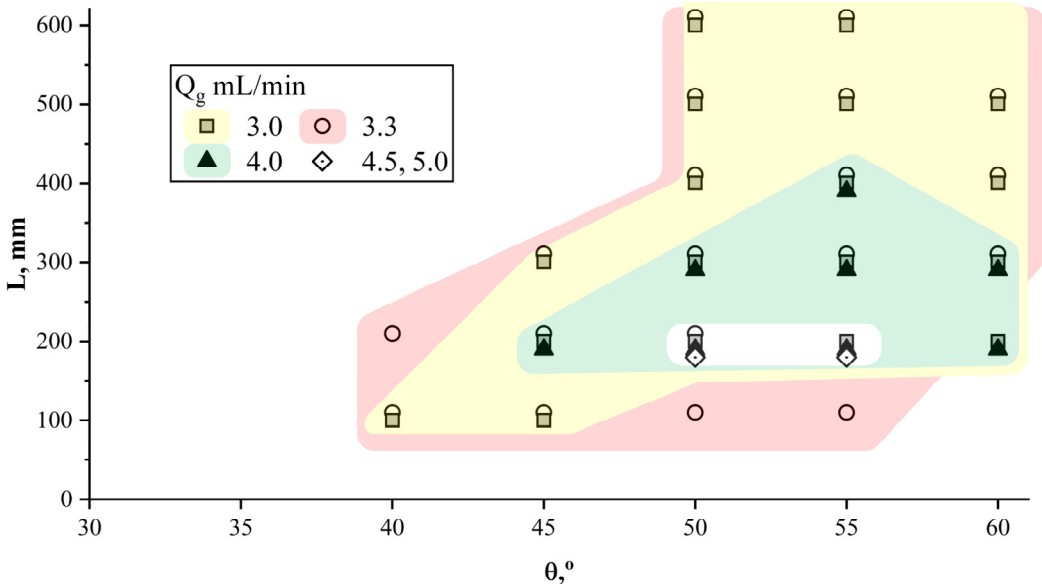

**Figure 6.** Map of the modes of movement of bubbles. The points correspond to the parameters for which the bubble chain mode appeared.

Chains of bubbles began to form at channel angles of 40° for gas flow rates 3.0 and 3.3 mL/min. Increasing the angle of inclination of the pipe increased the friction of the bubbles against the upper wall of the pipe. This led to a decrease in the transverse vibrations of the bubbles and allowed the chain of bubbles to form.

For gas flow rates of 3.0 and 3.3 mL/min, with an increase in the angle of inclination, the distance at which the bubble chain mode could be observed also increased up to 600 mm.

For gas flow rates of 4.0 mL/min, the maximum distance at which the bubble chain mode was observed first increased with an increase in the angle of inclination, and then decreased for channel inclination angles of more than 50°. For gas flow rates of 4.5 and 5.0 mL/min, clusters of bubbles were formed only at the distance of $L$ = 200 mm and at angles of 45° and 50°.

For gas flow rates of more than 5 mL/min, the bubble chain mode was not observed. The coalescence of bubbles near the capillary and those moving along the channel led to an increase in the diameter of individual bubbles, and the greater distance between individual bubbles did not allow them to form a chain.

### 3.2. Average Size of Gas Bubbles in the Inclined Tube

The dependence of the average size of gas bubbles as a function of the pipe inclination angle is shown in Figure 7.

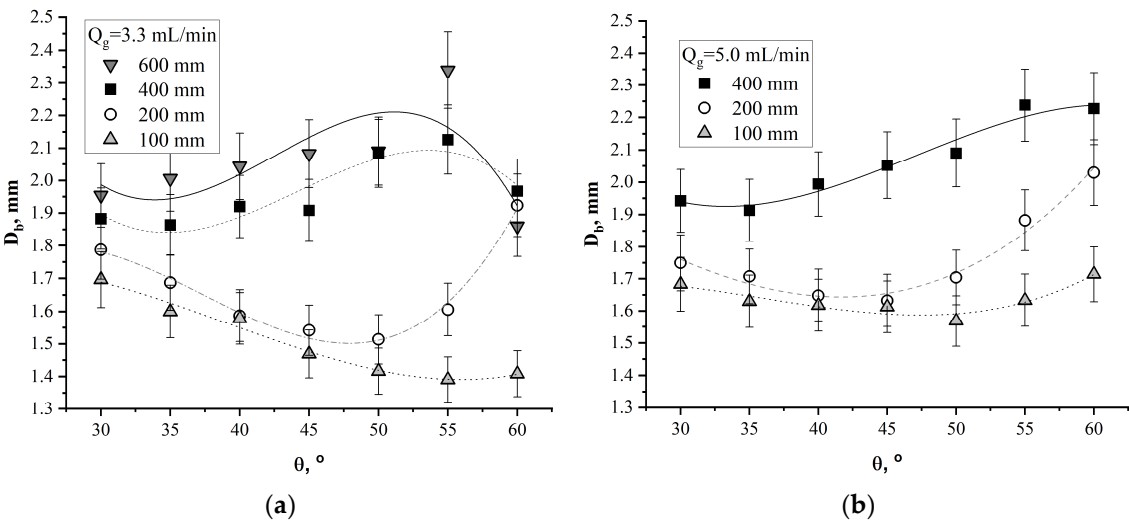

**Figure 7.** Dependence of the average size of gas bubbles as a function of the pipe inclination angle. Gas flow rate (**a**) $Q_g$ = 3.3 mL/min; (**b**) $Q_g$ = 5.0 mL/min.

The average size of the gas bubbles was $D_b$ = 1.4–2.3 mm for the gas flow rate $Q_g$ = 3.3 mL/min and $D_b$ = 1.5–2.2 mm for the gas flow rate $Q_g$ = 5.5 mL/min.

At a distance of 100 mm for the gas flow rate $Q_g$ = 3.3 mL/min (Figure 7a), the average diameter decreased with an increase in the angle of inclination of the pipe, since the departure diameter decreased due to the angle of inclination of the capillary. At a distance of 200 mm and pipe inclination angles up to 50°, the average diameter decreased due to the decrease in the departure diameter. However, at angles of inclination of more than 50°, bubbles were actively grouped when moving along the upper wall, and the average diameter increased due to coalescence. At distances of 400 and 600 mm, the effect of bubble coalescence along the upper wall of the inclined pipe compensated for the decrease in the departure diameter in the range of channel inclination angles from 30° to 50°. The average diameter of gas bubbles at the flow rate of 5 mL/min (Figure 7b) demonstrated similar behavior.

### 3.3. Average Velocity of Gas Bubbles in the Inclined Tube

The dependence of the average velocity of gas bubbles as a function of the pipe inclination angle is shown in Figure 8.

As the angle of inclination increased, the projection of the Archimedes force along the axis of movement of the bubbles decreased. Due to this, the bubbles were pressed against the upper wall of the inclined pipe and moved more slowly due to the friction generated by their movement against the wall.

At a distance of 100 mm for the gas flow rate $Q_g$ = 3.3 mL/min (Figure 8a), the velocity of the bubbles decreased throughout the selected range of angles. At a distance $L$ = 200 mm for angles of 55° and 60°, an increase in the velocity of the bubbles was observed. This was due to the fact that the bubbles were actively coalescing when moving along the upper wall of the pipe, and the average velocity of the bubbles increased due to the Archimedes force. The bubble chain mode for angles greater than 55 was not obtained. For the gas flow rate $Q_g$ = 5.0 mL/min (Figure 8b), the dependences of the average velocity of the bubbles for different angles of inclination of the pipe demonstrated similar behavior. An increase in the average velocity for the same distance L was associated with the increase in the average diameter of the bubbles as the gas flow rate increased. The nonlinearity in the average

velocity graphs for the distance $L = 400$ mm was associated with the coalescence of bubbles when moving along the upper wall of an inclined pipe. This was also typical for gas flow rate $Q_g = 3.3$ mL/min.

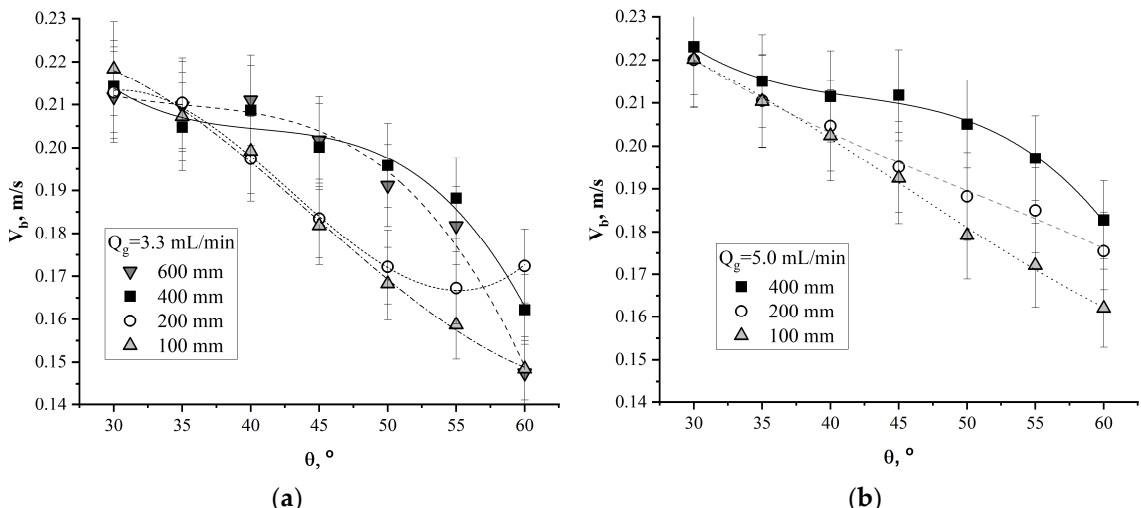

**Figure 8.** Dependence of the average velocity of gas bubbles as a function of the pipe inclination angle. Gas flow rate (**a**) $Q_g = 3.3$ mL/min; (**b**) $Q_g = 5.0$ mL/min.

The movement of bubbles was caused by buoyancy forces, surface tension, and the force of friction against the upper wall of the inclined channel. One of the most important dimensionless characteristics of the ascent of gas bubbles is the Reynolds number $Re_b = \rho V_b D_b / \mu$, where $\varrho$ is the density of the fluid, $\mu$ is the dynamic viscosity of the fluid, $V_b$ is the mean velocity of bubbles, and $D_b$ is the mean diameter of the bubbles. Figure 9 shows the values of the Reynolds number for gas bubbles, depending on the distance between the gas input point and the point of measurement.

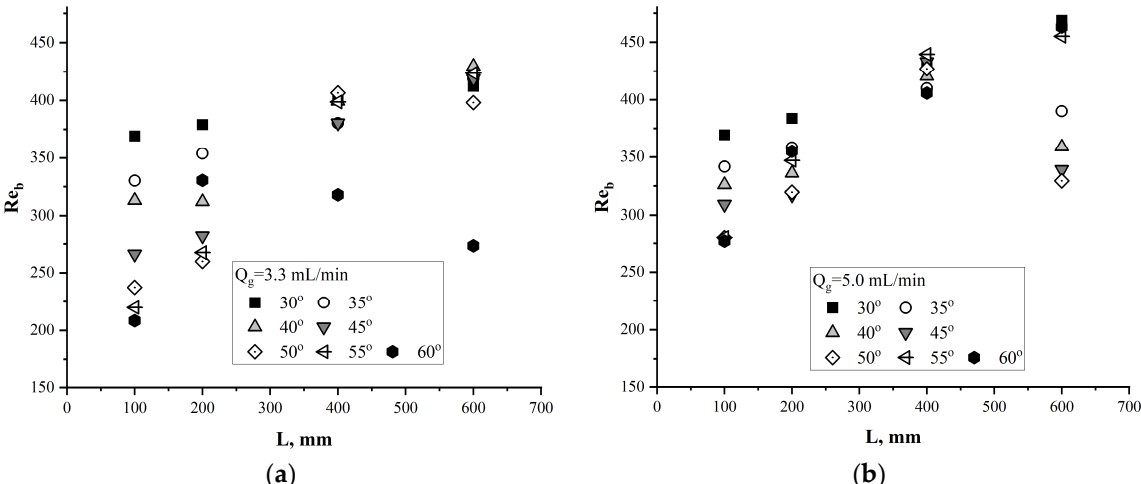

**Figure 9.** The dependence of the Reynolds number $Re_b$ on the distance between the gas input point and the camera shooting point. Angle of inclination of the pipe $\theta = 30°$–$60°$, gas flow rate (**a**) $Q_g = 3.3$ mL/min; (**b**) $Q_g = 5.0$ mL/min.

As the distance from the gas input point to the measurement point increased from 100 to 600 mm, the graphs for all pipe inclination angles (except 60°) tended toward the value of the Reynolds number $Re_b = 390$–450 for a gas flow rate $Q_g = 3.3$ mL/min (Figure 9a). The behavior of the Reynolds number at the channel angle of 60° differed significantly from its behavior at other angles. From this angle of inclination, the velocity of bubbles due to

coalescence and the growth of the average diameter were significantly reduced, leading to a decrease in the Reynolds number.

For the gas flow rate $Q_g$ = 5.0 mL/min (Figure 9b), the graphs tended toward the value of the Reynolds number $Re_b$ = 390–450 at a distance from the gas input point to the camera shooting point $L$ = 400 mm. With an increase in the distance to $L$ = 600 mm, due to coalescence, the differences in the diameters and velocity of the bubbles increased, leading to a discrepancy in the graphs of the Reynolds number.

## 4. Conclusions

An experimental study of bubble size from a single capillary in an inclined tube was presented.

A map of bubble movement modes was built. It is shown for which values of the gas flow rate, the angle of inclination of the pipe, and the distance from the gas input point to the measurement point the individual bubbles form a chain of bubbles. For gas flow rates of 3.0 and 3.3 mL/min, chains of bubbles were formed in a wide range of pipe inclination angles and distances from the gas input point to the shooting point. When the gas flow rate increased to 4.0 mL/min or more, the range of formation of chains of bubbles decreased due to the coalescence of bubbles as they moved along the upper wall of the inclined pipe. At gas flow rates of more than 5.0 mL/min, chains of bubbles were not formed.

The values of the average diameters were obtained depending on the angle of inclination of the pipe. The average diameter of the gas bubbles was $D_b$ = 1.4–2.3 mm for the gas flow rate $Q_g$ = 3.3 mL/min and $D_b$ = 1.5–2.2 mm for the gas flow rate $Q_g$ = 5.5 mL/min. With an increase in the angle of inclination of the channel, the average diameter of the gas bubbles decreased. The nonlinearity of the graphs of the average diameter was caused by the influence of the angle of inclination on the departure diameter, the coalescence of bubbles near the capillary, and the coalescence of bubbles when moving along the upper wall of the inclined pipe.

Dependences of the average velocity of bubbles on the angle of inclination of the pipe were obtained. With an increase in the angle of inclination of the pipe, the average velocity of the bubbles decreased. This was due to the friction on the upper wall and an increase in the influence of the Archimedes force. With an increase in the angle of inclination of the channel from 30° to 60°, the average velocity of bubbles decreased from ~0.22 m/s to ~0.15 m/s for the gas flow rate $Q_g$ = 3.3 mL/min and to ~0.18 m/s for the gas flow rate $Q_g$ = 5.0 mL/min. The nonlinearity in the average velocity graphs for distances $L$ = 400 mm and greater was associated with the coalescence of bubbles when moving along the upper wall of an inclined pipe.

The results can be used to verify CFD packages and for a deeper analysis of the results of complex gas–liquid flows, including under the conditions of nuclear reactors.

**Author Contributions:** Conceptualization, A.E.G., V.V.R., A.V.C. and O.N.K.; methodology, A.E.G., V.V.R. and A.V.C.; software, A.V.C.; investigation, A.E.G.; resources, A.E.G., A.V.C. and O.N.K.; data curation, A.E.G. and A.V.C.; writing—original draft preparation, A.E.G., A.V.C. and O.N.K.; writing—review and editing, A.E.G., A.V.C. and O.N.K.; project administration, O.N.K. All authors have read and agreed to the published version of the manuscript.

**Funding:** This research was funded by RSF, grant number 22-21-20029 and Ministry of Science and Innovation Policy of the Novosibirsk region, https://rscf.ru/project/22-21-20029/ (accessed on 27 January 2023).

**Institutional Review Board Statement:** Not applicable.

**Informed Consent Statement:** Not applicable.

**Conflicts of Interest:** The authors declare no conflict of interest.

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
