# Peer review of "The Effect of the Angle of Pipe Inclination on the Average Size and Velocity of Gas Bubbles Injected from a Capillary into a Liquid"

_water, doi:10.3390/w15030560_

Round 1

Reviewer 1 Report

In the article, an interesting problem of two-phase flow in an inclined capillary is studied experimentally. Depending on the inclination angle, the mean gas bubble diameter and the gas bubble velocity at different gas volume flows of the shadow photography method are investigated. My comments to the article are as follows:

Chapter 1: 

Please mention at the beginning in the motivation section application cases in which this type of two-phase flow is relevant. How great is the influence, for example, in the area of heat transfer.

line 29: "Weber number" (insert "number"), delete "number" after "Morton number" instead.

line 38: insert “are” between “… channels widely …”

line 64: “on or near the wall” instead of “near or 

line 74: dimension “mm3” instead of “mm”

lines 83/84: “… (CFD, Computational Fluid Dynamics).” instead of “… (CDF, Computational fluid dynamics).”

line: 89: “…, Direct Numerical Simulation (DNS), …” instead of “…, direct numerical modeling (DNS), …”

line 95: use “validation” instead of “verification”.

Chapter 2:

Table 1: The value for the dynamic viscosity of water at 20°C is wrong: 1 mPas instead of 1000 mPas

Equation (1) and (2): Please mention what the variable S is.

Equation (3): Why do you assume a behavior like that of a solid sphere? How great is the influence of an immobilized surface due to the presence of surfactants? Distilled water has been used after all.

Figure 3: The use of error bars would be very helpful here. Also, please mention that this is the Reynolds number based on the bubble diameter.

Chapter 3:

Please change the chapter name from "Results and Discuss" to "Results and Discussion".

Figures 6, 7, 8 and 9: How many measurements were taken at each operating point? How good is the reproducibility? Over how many bubbles was the mean bubble diameter and the mean gas bubble velocity averaged?

Line 248: “Figure 9” instead of “Figure 8”.

Lines 231, 237, 258: Please insert a space between "Fig." and the number

Chapter 4: At the end of the chapter, please describe how the findings from these studies and investigations can be applied to apparatus design, for example. What are the advantages of the present findings?

References: For references 8, 12, 15, and 17, please list all authors instead of "et al." - as for the other references with more than two authors.

Author Response

Thank you for your review!

Point 1: Please mention at the beginning in the motivation section application cases in which this type of two-phase flow is relevant. How great is the influence, for example, in the area of heat transfer.

Response 1: Thank you for your attention to this subject. The motivation section has been expanded, works on the study of heat transfer in downflows and in pipes with sudden extensions have been considered. The paper [1.  Lobanov, P.D. Wall Shear Stress and Heat Transfer of Downward Bubbly Flow at Low Flow Rates of Liquid and Gas. J. Eng. Thermophys. 2018, 27, 232–244. https://doi.org/10.1134/S1810232818020091] shown that in downward bubbly flow change in the size of the dispersed phase can lead to both intensification and deterioration of heat transfer as compared with a single-phase flow at constant flow rates of liquid and gas at the channel inlet. Adding small gas bubbles to a flow leads to “laminarization” in the wall region and deterioration in the heat transfer by about 25% as compared with a single-phase flow. Large bubbles lead to higher turbulence level in the near-wall region, increase in the average friction, and intensification of the heat transfer by up to 50%. In the paper [2.               Lobanov, P.; Pakhomov, M.; Terekhov, V. Experimental and Numerical Study of the Flow and Heat Transfer in a Bubbly Turbulent Flow in a Pipe with Sudden Expansion. Energies 2019, 12, 2735. https://doi.org/10.3390/en12142735], the addition of air bubbles led to a significant increase in the heat transfer rate (up to 300%) in a downstream bubbly flow in a sudden pipe expansionare. (line 20-29)

Point 2: line 29: "Weber number" (insert "number"), delete "number" after "Morton number" instead.

line 38: insert “are” between “… channels widely …”

line 64: “on or near the wall” instead of “near or

line 74: dimension “mm3” instead of “mm”

lines 83/84: “… (CFD, Computational Fluid Dynamics).” instead of “… (CDF, Computational fluid dynamics).”

line: 89: “…, Direct Numerical Simulation (DNS), …” instead of “…, direct numerical modeling (DNS), …”

line 95: use “validation” instead of “verification”.

Table 1: The value for the dynamic viscosity of water at 20°C is wrong: 1 mPas instead of 1000 mPas

Response 2: Thank you for your comments very much. The errors has been corrected (line 39, 46, 69, 79, 88, 94, 100, table 1)

Point 3: Equation (1) and (2): Please mention what the variable S is.

Response 3: The mention of the variable S (the area of the bubble on the image) is included in the text (line 144)

Point 4: Equation (3): Why do you assume a behavior like that of a solid sphere? How great is the influence of an immobilized surface due to the presence of surfactants? Distilled water has been used after all.

Response 4: Thank you for your attention to this subject. Optical observations, including with the help of high-speed cameras (the shooting speed is more than 1000 frames per second) in experiments on the study of gas-liquid flows in an inclined flat channel showed a absenceof shape pulsations for bubbles of small diameters (up to 3-4 mm).

The appearance of even a small amount of surfactants and pollutants led to a sharp decrease in the departure diameter of the bubbles and prevented coalescence.

Point 5: Figure 3: The use of error bars would be very helpful here. Also, please mention that this is the Reynolds number based on the bubble diameter.

Response 5: Thank you for your comments very much. Error bars have been added, it was stated that Reynolds number Reb calculated relative to the velocity Ub and diameter Db of the gas bubble (line

Point 6: Please change the chapter name from "Results and Discuss" to "Results and Discussion".

Response 6: Thank you for your comments very much. The error has been corrected.

Point 7: Figures 6, 7, 8 and 9: How many measurements were taken at each operating point? How good is the reproducibility? Over how many bubbles was the mean bubble diameter and the mean gas bubble velocity averaged?

Response 7: Thank you for your question. At least 1000 bubbles were processed for each point to determine the average diameter and velocity. For modes without coalescence, the overwhelming number of bubbles has a diameter close to the value of the tear-off diameter. For modes with coalescence, the average diameter lies in the range between the departure diameter and the diameter of bubbles with the doubled volume.

The repeatability of the results largely depends on the purity of the liquid, so after a series of experiments, the water in the working volume was replaced. Even small concentrations of surfactants or contaminants had a significant effect on the diameter distribution, the average diameter of gas bubbles and, accordingly, the velocity.

Point 8: Line 248: “Figure 9” instead of “Figure 8”.

Lines 231, 237, 258: Please insert a space between "Fig." and the number

Response 8: Thank you for your comments very much. The errors has been corrected.

Point 9: Chapter 4: At the end of the chapter, please describe how the findings from these studies and investigations can be applied to apparatus design, for example. What are the advantages of the present findings?

Response 9: The results can be used to verify CFD pacages, and for a deeper analysis of the re-sults of complex gas-liquid flows, including in the conditions of nuclear reactors.

Point 10: References: For references 8, 12, 15, and 17, please list all authors instead of "et al." - as for the other references with more than two authors.

Response 10: Thank you for your comments very much. Errors in the citation style have been corrected.

Reviewer 2 Report

Considering the remarks below, I cannot recommend the manuscript for publication in the present form. The results seem to need a thorough revision. The explanations for the effects, especially concerning the droplet diameters, are very speculative. The presented results seem to be partly inconsistent. 

Equation 2, derivation is not clear. See attached pdf.

Fig 3: The link to ref 22 in figure 3 is not clear. In ref 22 the drift velocity is not explicitly presented as function of the Re-number. Moreover, it is unusual to represent a dimensional quantity (Ub) as a function of a dimensionless quantity (Reynolds). Additionally, in ref 22 apparently experiments were conducted for larger bubbles, compared to tube diameter. Is this really comparable to the present exp. setup.

The bubble Re-number in fig 3 ranges from 300-500. In Fig 9,  typical values are well below 1 for various inclinations. This seems to be inconsistent. Please comment.

Fig 4: picture for theta=50° appears twice?

Fig. 7. Please explain the effect of inclination and initial bubble diameter. For a vertical pipe (theta=0) the hydrostatic pressure at the injection point should be higher than for an inclined pipe. This should cause a smaller bubble diameter for small theta. However, the present experimental results show the contrary trend. The fact, that bubble diameters are tending to increase with length (=height) is comprehensible.

For theta>50°, why is there an increase of dp(200mm) but a decerase of dp for 400 and 600mm?

Author Response

Thank you for your review!

Reviewer 3 Report

The work under review consists of an experimental study on the effect of coalescence on the average diameter and velocity of gas bubbles in an inclined pipe. In general, the work is interesting. However, the novelty needs to be pointed out and more detail on the methodology and accuracy obtained is also needed, in order for the work to be suitable for publication. Bellow are some comments that are proposed to the Authors to improve the manuscript.

Line 29-30: please rephrase.

Line 44: Better use the word “approximately”. This to be substituted throughout the manuscript.

Line 64: typo “…near and near…”. Please modify accordingly

Line 83: typo “CDF”. Please substitute with the correct CFD.

INTRODUCTION: the novelty of the work needs to be explicitly described.

Line 113: Please substitute the number 1 with the word “one”

Table 1: 3 has to be a superscript.

Formulae 1 and 2. Please provide the description of the notations.

Figure 2: Details about both the software and the procedure are needed. Please provide a detailed discussion on that.

Line 144: The Authors need to provide a justification on why the accuracy is 1 pixel. How has this been determined?

Section 2: More detail about the method and the accuracy is needed.

Line 157: Please modify the title :…and Discussion”

Figure 4: A better representation is needed, in order for an efficient comparison to be conducted. For example, the scale, distances, direction of the flow etc.

Figure 5: For theta 35 degrees it shows that there are clusters and for theta 50 degrees a chain is formed.  In figure’s caption it is stated: “Bubble movement in modes without cluster formation (θ = 35O) and with cluster formation (θ = 50O), gas flow rate Qg = 3.3 ml/min.” A better description of what is a cluster formation and what is a chain formation is needed.

Additionally, a criterion that distinguishes the modes is needed to be set by the Authors(e.g., similar distance between how many bubbles? What would be the acceptable range that a “similar” distance will be valid etc).

Line 216: Please add “to” after “up”.

Figure 7b and 8b: Please provide the reason why there are no data given for L=600mm?

Line 225: is

Line 247: Please modify the phrase “…where ρ are the density of the fluid, μ are the dynamic viscosity of the fluid, 247 Vand Dare velocity and diameter of bubbles.” There are typos and grammar issues. Additionally, details of the diameter of bubbles are needed- is it the mean diameter?

General Comment: How many repetitions have been conducted? What is the accuracy of the experimental results? This information has to be depicted in the analysis and the diagrams that are provided in the manuscript. Error bars to all diagrams are needed.

In summary, the work under review could be published after the Authors go through a thorough revision according to the proposed comments and corrections.

Author Response

Thank you for your review!

Point 1: the novelty of the work needs to be explicitly described.

Response 1: Thank you for your attention to this subject.

This work is devoted to the experimental study of the effect of coalescence on the average size and velocity of gas bubbles in an inclined pipe. The values of the average diameters and average velocities of bubbles were obtained depending on the angle of inclination of the pipe. A map of regime parameters was constructed at which gas bubbles form a stable structure – a chain of bubbles with an equal diameter. There is a lack of experimental data in this area, since the main attention is paid to small angles of inclination of pipes and channels.

Point 2: Line 113: Please substitute the number 1 with the word “one”

Table 1: 3 has to be a superscript.

Response 2: Thank you for your comments very much. The errors has been corrected.

Point 3: Formulae 1 and 2. Please provide the description of the notations.

Response 3: Thank you for your attention to this subject. The description of the notations were included to the text

Point 4: Figure 2: Details about both the software and the procedure are needed. Please provide a detailed discussion on that.

Response 4: Thank you for your attention to this subject. The processing was carried out in two stages. At the first stage, the videos were split into separate frames and translated into the "Grayscale" format. Аaccording to the level of illumination and the gradient of illumination, the boundaries of objects were found and frames were binarized. At the second stage of processing, the properties of objects are found – size, location. Сorrelation analysis of successive frames was carried out to determine the speed and trajectories of bubbles.

Point 5: Line 144: The Authors need to provide a justification on why the accuracy is 1 pixel. How has this been determined?

Response 5: Thank you for your question. The inaccuracy of the wording has been corrected. After processing and binarization of images, еhe accuracy of determining the position of the bubble boundary was ± 1 pixel, so, depending on the size of the bubble, the relative error ranged from 0.01 to 0.1 for bubbles from 0.3 to 7 mm in size.

Point 6: Section 2: More detail about the method and the accuracy is needed.

Response 6: Thank you for your attention to this subject. In the text, the additional influence of the error of the devices used was given. The part on checking the measuring system has been expanded.

Point 7: Line 157: Please modify the title :…and Discussion”

Line 216: Please add “to” after “up”.

Line 225: is

Response 7: Thank you for your comments very much. The errors has been corrected.

Point 8: Figure 4: A better representation is needed, in order for an efficient comparison to be conducted. For example, the scale, distances, direction of the flow etc.

Response 8: Thank you for your comments very much. The direction of movement of the bubbles and the scale were indicated in the figure.

Point 9: Figure 5: For theta 35 degrees it shows that there are clusters and for theta 50 degrees a chain is formed.  In figure’s caption it is stated: “Bubble movement in modes without cluster formation (θ = 35O) and with cluster formation (θ = 50O), gas flow rate Qg = 3.3 ml/min.” A better description of what is a cluster formation and what is a chain formation is needed.

Additionally, a criterion that distinguishes the modes is needed to be set by the Authors(e.g., similar distance between how many bubbles? What would be the acceptable range that a “similar” distance will be valid etc).

Response 9: Thank you for your comments. The paper talks about the formation of a cluster-a chain of bubbles when the following conditions are met:

  • Bubbles of the same diameter
  • The same speeds of individual bubbles

For these modes of bubble movement, it is characteristic that the average values of bubble size and velocity are close to the values of size and velocity for a single bubble in a cluster (deviation of no more than 5%).

For a 35-degree pipe angle, the formation of a cluster chain does not occur, since there are several typical bubble diameters in this mode.

Point 10: Figure 7b and 8b: Please provide the reason why there are no data given for L=600mm?

Response 10: Thank you for your attention to this subject. At high gas flow rates (5 ml/min or more) and large distances from the gas inlet to the optical section (600 mm or more), the distribution of bubble diameters and the average diameter of gas bubbles were largely random (since coalescence and crushing of bubbles are probabilistic). Under such conditions, the repeatability of the experiments was lower than the reference values. Further studies are required for these parameters.

Point 11: Line 247: Please modify the phrase “…where ρ are the density of the fluid, μ are the dynamic viscosity of the fluid, 247 Vb and Db are velocity and diameter of bubbles.” There are typos and grammar issues. Additionally, details of the diameter of bubbles are needed- is it the mean diameter?

Response 11: Thank you for your comments very much. The errors has been corrected.

Point 12: How many repetitions have been conducted? What is the accuracy of the experimental results? This information has to be depicted in the analysis and the diagrams that are provided in the manuscript. Error bars to all diagrams are needed.

Response 12: At least 1000 bubbles were processed for each point to determine the average diameter and velocity. The errors are indicated in the installation and methodology section and are reflected in the diagrams

Round 2

Reviewer 2 Report

Enclosed some minor corrections.

Line 30:       expansionare.

Line 49.       It was shown that the presence various values of total dissolved solid materials can affect the thermal conductivity of water and the bubble formation characteristics.  Check grammar

Author Response

The team of authors thanks the reviewer for comments.

Errors have been corrected (line 30, 49). The manuscript was checked for other grammatical errors.

Reviewer 3 Report

The Authors have addressed most of the proposed comments and suggestions.

Author Response

The team of authors thanks the reviewer for questions and comments.